



# Monoterpene oxidation pathways initiated by acyl peroxy radical addition

Dominika Pasik[1,2], Thomas Golin Almeida[1,2], Emelda Ahongshangbam[1,2], Siddharth Iyer[3], and Nanna Myllys[1,2]

[1]Department of Chemistry, University of Helsinki, Helsinki, 00014, Finland
[2]Institute for Atmospheric and Earth System Research, University of Helsinki, Helsinki, 00014, Finland
[3]Aerosol Physics Laboratory, Tampere University, Tampere, 33014, Finland

**Correspondence:** Dominika Pasik (dominika.pasik@helsinki.fi) and Nanna Myllys (nanna.myllys@helsinki.fi)

**Abstract.** Monoterpenes are released into the atmosphere in significant quantities, where they undergo various oxidation reactions. Despite extensive studies in this area, there are still gaps that need to be addressed to fully understand the oxidation processes occurring in the atmosphere. Recent findings suggest that reactions between alkenes and acyl peroxy radicals can be competitive under atmospheric conditions. In this study, we investigate the oxidation reactions of seven monoterpenes with

acyl peroxy radicals and the subsequent diverse reactions, including accretion, alkyl radical rearrangements, H-shift, and $\beta$-scissions reactions. The accretion reaction leads to the release of excess energy, which is sufficient to open small secondary rings in the alkyl radical structures. This reaction is most significant for sabinene (39%) and $\alpha$-thujene (20%). A competing reaction is $O_2$ addition, which the majority of alkyl radicals undergo, subsequently leading to the formation of peroxy radicals. These then react further, forming alkoxy radicals that can subsequently undergo $\beta$-scission reactions, among other fates . Our

calculations show that for $\beta$-pinene, camphene, and sabinene, which have an exocyclic double bond, $\beta$-scission rearrangements result in radicals capable of undergoing further propagation of the oxidative chain, while for limonene, $\alpha$-pinene, and $\alpha$-thujene, scissions leading to closed-shell products that terminate the oxidative chain are preferred. The results indicate that if reactions of monoterpenes with acyl peroxy radicals are indeed competitive under atmospheric conditions, their oxidation would lead to more oxygenated compounds with a higher molar mass, potentially contributing to secondary organic aerosol yields. Moreover,

this study highlights the significance of stereochemistry in controlling the oxidation of monoterpenes initiated by acyl peroxy radicals.

## 1 Introduction

Monoterpenes ($C_{10}H_{16}$) represent a diverse group of compounds emitted by plants and trees, comprising roughly 15% of biogenic volatile organic compound emissions into the atmosphere (Friedman and Farmer, 2018; Hantschke et al., 2021; Guenther

et al., 2012; Lun et al., 2020). Due to their high reactivity, they undergo numerous oxidation reactions in the atmosphere, which can result in compounds with higher molar mass and lower volatility (Iyer et al., 2021, 2018; Kanakidou et al., 2005; Boyd et al., 2015; Bianchi et al., 2019; Zhao et al., 2015; Vereecken and Nozière, 2020). These in turn, contribute to the production of secondary organic aerosols (SOA), which have significant impacts on climate, air quality, and human health (Friedman and



Farmer, 2018; Ehn et al., 2014; Xu et al., 2015, 2018). While it is well established that aerosol particles play a crucial role in the atmosphere by influencing Earth's radiative balance and serving as cloud condensation nuclei (CCN), and extensive studies have been conducted to understand their effects, there still exists high uncertainty in global climate models, largely due to the limited understanding of SOA formation from biogenic VOCs (Glasius and Goldstein, 2016; Jokinen et al., 2012; Shrivastava et al., 2017; Dada et al., 2023). Numerous studies estimate that monoterpenes contribute to approximately half of the global biogenic SOA source, primarily through reactions with OH radical, ozone, and $NO_3$ radical (Spracklen et al., 2011; Day et al., 2022; Zhang et al., 2018; Presto et al., 2005). The oxidized organic compounds formed in these reactions have a lower volatility compared to their parent monoterpene, and thus can participate in aerosol particle formation or growth by either creating new particles through homogeneous nucleation or increasing the size of existing newly formed particles through condensation.

Despite their structural similarities, monoterpenes exhibit different oxidation reactivity, and thus different SOA yields. For instance, Lee et al. (2006) showed that the ozonolysis of monoterpenes resulted in SOA yields ranging from 17% for $\beta$-pinene to 41% for $\alpha$-pinene. Moreover, the trends in SOA yields vary depending on the oxidant (Draper et al., 2015; Liu et al., 2022). Recently, Draper et al. (2024) investigated the reactions of monoterpenes with $NO_3$ followed by ring-opening reactions and bond scissions. Their research not only provided evidence that each monoterpene possesses a unique reactivity towards the same oxidation pathways but also presented rearrangement reactions that open up possibilities for further oxidation, thereby contributing to SOA yields. Their calculations indicate that the alkyl radical rearrangement outcompetes $O_2$-addition in the case of sabinene, while it is minor but competitive in the case of $\alpha$-thujene and $\beta$-pinene, and negligible in the case of camphene. These ring-opening rearrangements can promote further propagation of the oxidative chain.

Peroxy radical ($RO_2$) reactions with VOCs were initially thought to be negligible at room temperature and were thus overlooked in atmospheric chemistry. However, recent studies by Nozière and Fache (2021) revealed that unsaturated hydrocarbons can indeed react with $RO_2$, and reaction was initially thought to lead epoxides like in combustion conditions. Later Nozière et al. (2023) continued kinetic experiments of $RO_2$ with unsaturated hydrocarbons, and suggested an accretion pathway for this reaction. Additionally, Pasik et al. (2024a) demonstrated that reactions between isoprene and monoterpenes with acyl peroxy radicals (APRs) occur at rates significant enough to impact atmospheric chemistry. For example, reaction of limonene with $CH_3C(O)OO\cdot$ is fast enough to compete in the atmosphere, potentially accounting for up to 0.1% of the limonene sink. Moreover, monoterpene + APR reactions result in the formation of new and more functionalized products as in addition of oxidation reactions, which is a sequential and multi-step process of intra-molecular reaction followed by $O_2$ addition. Furthermore, oxidation by APR also increases the total number of carbon atoms and this, along with autoxidation, leads to lower-volatility compounds. APRs as novel oxidants open up numerous new reaction pathways, potentially elucidating missing data regarding SOA formation.

Building upon the research conducted by Draper et al. (2024) and Pasik et al. (2024a), in this study, we investigate the oxidation pathways of monoterpenes + APR through alkyl ring-opening reactions and alkoxy radical bond scissions. Utilizing computational methods, we explore these reactions in the $CH_3C(O)OO\cdot$-initiated oxidation of seven monoterpenes: limonene, $\alpha$-pinene, $\beta$-pinene, camphene, sabinene, $\alpha$-thujene, and $\Delta^3$-carene (see Figure 1). This research aims to solve the processes





leading to oxygenated multifunctional organic compounds possibly contributing to the formation of secondary organic aerosols. Understanding these processes is crucial for addressing the enduring knowledge gap in the formation of new particles.

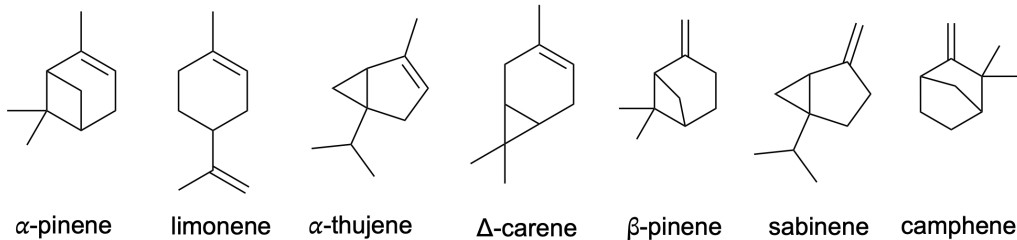

**Figure 1.** Structures of investigated monoterpenes.

## 2   Methods

### 2.1   Computational details

To investigate the accretion reactions between APRs and monoterpenes as well as subsequent bond-scission reactions, the first step involved locating the reactants, products, and corresponding transition states using density functional theory (DFT) at the $\omega$B97X-D/6-31+G* level of theory (Pasik et al., 2024b; Chai and Head-Gordon, 2008; Myllys et al., 2016b; Hariharan and Pople, 1973). These served as input for conformational sampling, which was carried out using Conformer–Rotamer Ensemble Sampling Tool (CREST) software at the GFN2-xTB level (Pracht et al., 2020; Bannwarth et al., 2019). For sabinene, camphene, $\Delta$-carene, and $\alpha$-thujene, only the dominant APR addition pathway leading to the tertiary alkyl radical was taken into account. For $\beta$-pinene, and limonene, we utilized the results published in our previous work (Pasik et al., 2024a). For $\alpha$-pinene, a comprehensive analysis was conducted (see SI for details). Since the electronic energies obtained at the xTB level correlate well with DFT energies, we further optimized conformers within 2.5 kcal/mol relative to the lowest-energy conformer. For reactants and products, the reduced number of structures were optimized at the $\omega$B97X-D/6-31+G* level of theory. Based on electronic energy, dipole moment and squared rotational constant values, duplicate structures were removed, and only unique conformers were considered. The generated conformers for transition state (TS) structures were optimized at the DFT level, while maintaining the breaking/created bond distance frozen. Following this, the full TS optimization and frequency calculation was conducted. Transition states were verified by the presence of exactly one imaginary frequency. To calculate the reaction energy barrier more accurately, on top of the DFT structures linear-scaling coupled-cluster with single and double excitations as well as perturbative-inclusion-of-triples single-point calculations were performed using the DLPNO-CCSD(T)/aug-cc-pVTZ method (Riplinger et al., 2013, 2016; Myllys et al., 2016a; Draper et al., 2019; Dunning Jr, 1989). For all conformers, the zero-point corrected energies were calculated. The barrier heights were computed as the difference between the energies of the lowest energy conformers of TS and the reactants.



The reaction rate coefficients were calculated using the multiconformer transition state theory (MC-TST) approach for bimolecular reactions with the following expression (Vereecken and Peeters, 2003; Viegas, 2023):

$$k_{\text{bi}} = \kappa_t \frac{k_\text{B}T}{hP_{ref}Q_{\text{MT}}} \frac{\sum_i^{\text{allTSconf.}} \exp(-\frac{\Delta E_i}{k_\text{B}T})Q_{\text{TS},i}}{\sum_j^{\text{allRconf.}} \exp(-\frac{\Delta E_j}{k_\text{B}T})Q_{\text{R},j}} \exp(-\frac{E_{\text{TS}} - E_{\text{R}}}{k_\text{B}T}), \tag{1}$$

where $\kappa_t$ is quantum-mechanical tunneling coefficient ($\kappa_t$=1 is used for the reactions involving atoms other than hydrogen), $T$ is the temperature (=298.15 K), $h$ is the Planck's constant, $P_{\text{ref}}$ is the reference pressure (=1atm, here we used $2.45{\times}10^{19}$ cm$^{-3}$ value) and $k_\text{B}$ is the Boltzmann's constant. $Q_{R,j}$ and $Q_{TS,i}$ are the partition functions of the reactant (acylperoxyl radical) and transition state conformers, respectively, $Q_{MT}$ is a partition function for the monoterpene. $\Delta E_j$ and $\Delta E_i$ correspond to the zero-point corrected electronic energies of the reactant and transition state conformers relative to the lowest-energy conformers, respectively. $E_R$ and $E_{TS}$ are the zero-point corrected electronic energies of the lowest energy reactant and transition state conformers. The partition functions were calculated at the $\omega$B97X-D/6-31+G* level of theory, while the reaction barrier ($E_{TS} - E_R$) includes the DLPNO-CCSD(T)/aug-cc-pVTZ energy correction.

For the hydrogen shift reactions and monoterpene + APR-derived alkoxy radical bond scissions, the unimolecular reaction rate coefficients were calculated following the equation (Møller et al., 2016; Zhao et al., 2022):

$$k_{\text{uni}} = \kappa_t \frac{k_\text{B}T}{h} \frac{\sum_i^{allTSconf.} \exp(-\frac{\Delta E_i}{k_\text{B}T})Q_{TS,i}}{\sum_j^{allRconf.} \exp(-\frac{\Delta E_j}{k_\text{B}T})Q_{R,j}} \exp(-\frac{E_{TS} - E_R}{k_BT}) \tag{2}$$

Additionally, IRC calculations were performed to ensure that the transition states are connected to the correct reactant and product wells. To calculate the tunneling coefficient $\kappa_t$, we utilized the Eckart tunneling method. This approach involves a one-dimensional calculation in which the tunneling coefficient is obtained by solving a three-parameter equation for an asymmetric one-dimensional potential, referred to as the Eckart potential. The calculation requires the transition state energy, along with the energies of the reactant and product connected to the lowest transition state structure, as well as the imaginary frequency. The Eckart tunneling method has been successfully applied in previous studies to calculate hydrogen shift reaction rates in similar systems (Hasan et al., 2020; Zhang and Dibble, 2011; Skodje et al., 1981).

To calculate alkyl radical ring-opening rearrangement we adopt the methodology outlined in the work by Draper et al. (2024). For the bicyclic monoterpenes where the 3-, 4- or 5-membered ring-opening rearrangement is possible ($\alpha$-pinene, $\alpha$-thujene, sabinene, $\beta$-pinene and camphene), the reaction dynamics were simulated using the Master Equation Solver for Multi Energy-well Reactions (MESMER) software (Glowacki et al., 2012). Vibrational frequencies and rotational constants were calculated at the DFT level while the zero-point corrected energies were calculated at the DLPNO-CCSD(T)/aug-cc-pVTZ//$\omega$B97X-D/6-31+G* level. The reactions in question were modeled using following methods: the APR-addition reaction was simulated using the Inverse Laplace Transform (ILT) method, assuming a barrierless association and using realistic bimolecular rate coefficients. O$_2$-addition reactions were attributed to the initial APR–alkyl radical and its ring-opening product, treating them as bimolecular sinks. The ring-break reactions, whose TS energetics were calculated, were modeled using Simple Rice-Ramsperger-Kassel-Marcus (RRKM) theory with Eckart tunneling correction. Bimolecular rate coefficients used were $1{\times}10^{-12}$ cm$^3$ molecule$^{-1}$ s$^{-1}$ for O$_2$ addition with O$_2$ concentration of $5.16{\times}10^{18}$ molecules cm$^{-3}$ (0.21 atm), and the





respective accretion reaction rate coefficients (used as pre-exponential factor in ILT model) for each monoterpene are listed
in Table 1 ($k_{TST-1}$). Modeling of the ring-opening reaction was conducted according to the procedure described by Draper
et al. (2024). We used exponential-down parameter $\Delta E_{down}$ = 225 cm$^{-1}$ and for each structure we estimated Lennard-Jones
parameters based on Joback and Reid (1987) work. N$_2$ was assigned as the bath gas with Lennard-Jones parameters $\epsilon/k_B$ =
91.85 K and $\sigma$ = 3.919 Å. We used grain size of 50 cm$^{-1}$ and the span of grains above the highest-energy transition state value
was 50 k$_B$T (see SI for full analysis). The concentration of CH$_3$C(O)OO· used was 10$^{15}$ cm$^{-3}$ to ensure rapid formation of the
modeled monoterpene-APR intermediate (RO-APR). The Lennard-Jones parameter values were estimated using the procedure
described by Gao et al. (2016) and Tee et al. (1966) based on the group-additivity method for deriving pure-compound critical
properties by Joback and Reid (1987) and are available in Supporting Information.

## 3 Results

### 3.1 Monoterpene-derived alkyl ring-opening reactions

In the first step, we considered the reaction between CH$_3$C(O)OO· and selected monoterpenes (see Figure 2). In our previous
study, we calculated reaction barriers for APR + limonene, $\alpha$-pinene, and $\beta$-pinene (Pasik et al., 2024a) according to which,
of the two possible addition sites, APR addition leading to a secondary alkyl radical was preferred for $\alpha$-pinene. Since we
expect that the formation of tertiary radicals should be faster, the analysis for $\alpha$-pinene was repeated, taking into account the
stereochemistry of the radical addition to the double bond. Our results show that this has a significant impact on the kinetics
of the reaction, and the addition of APR from the opposite side of the secondary ring and methyl groups (R-isomer) proceeds
with a significantly lower barrier of 1 kcal/mol and a rate of 2.6×10$^{-16}$ cm$^3$/s than if the attack occurs from the same side as
the methyl groups (S-isomer, barrier showed by Pasik et al. (2024a)). Detailed analysis is described in Supporting Information.
These new findings underscore the impact of stereochemistry on the reactions between monoterpenes and peroxy radicals.

Reaction barriers and calculated rate coefficients for APR + limonene and $\beta$-pinene (Pasik et al., 2024a) were adopted to this
study. All barrier heights and calculated reaction rate coefficients are summarized in Table 1. As can be expected based on our
previous study (Pasik et al., 2024a), all reaction barriers were low (0.2–3.1 kcal/mol), and the reaction rate coefficients were on
the order of 10$^{-15}$–10$^{-17}$ cm$^3$/s. The reaction rate coefficients for these reactions are sufficiently high to be considered feasible
under atmospheric conditions with realistic APRs concentrations. Similarly to $\alpha$-pinene, for the stereoisomers of $\alpha$-thujene,
the results vary significantly, indicating a structural dependency on reactivity. According to Table 1, the addition of APR to
the $\alpha$-thujene resulting in R-acetyl product isomer proceeds about one order of magnitude faster than for the S-acetyl product
isomer.

The resulting alkyl radical can react with molecular oxygen which, due to its high concentration in the atmosphere, is
considered the primary pathway for VOC oxidation. However, the addition of APR to the double bond in monoterpene, as
indicated by Table 1, is an exothermic process. The energy released in this reaction can contribute to overcoming the barrier for
a subsequent unimolecular ring-opening reaction, a reaction pathway that can lead to highly functionalized oxidized products.
Table 1 highlights the importance of considering energy released in reaction (excess energy, $\Delta E_{ex}$). There is a noticeable







**Figure 2.** Monoterpene + APR reactions. R-O stands for ring-opening rearrangements.

**Table 1.** Zero-point corrected energies [kcal/mol] for monoterpene + APR accretion reaction barrier ($\Delta E_1$) and its excess energy ($\Delta E_{ex}$) followed by ring-opening reaction ($\Delta E_2$) calculated at DLPNO-CCSD(T)/aug-cc-pVTZ//$\omega$B97X-D/6-31+G*. Corresponding reaction rate coefficients for monoterpene + APR accretion reaction ($k_{TST-1}$) [cm$^3$ s$^{-1}$] as well as ring-opening reaction calculated at the MC-TST ($k_{TST-2}$ [s$^{-1}$]) and RRKM-ME (($k_{RRKM}$) [s$^{-1}$]. Yield [%] of ring-opening products calculated with MC-TST (%$_{TST-2}$) and RRKM-ME (%$_{RRKM}$). RRKM-ME yields were taken from time profiles and MC-TST yields were calculated assuming a rate coefficient of $10^{-12}$ cm$^3$ molecule$^{-1}$ s$^{-1}$ with O$_2$ concentration of 5.16×10$^{18}$ molecules cm$^{-3}$ for the competing RO-APR + O$_2$ reaction. Excess energy ($\Delta E_{ex}$) [kcal/mol] calculated as difference between reactant and APR-monoterpene adduct energies.

| monoterpene | $\Delta E_1$ | $k_{TST-1}$ | $\Delta E_{ex}$ | $\Delta E_2$ | $k_{TST-2}$ | $k_{RRKM}$ | %$_{RRKM}$ | %$_{TST-2}$ |
|---|---|---|---|---|---|---|---|---|
| $\beta$-pinene | 2.4 | 2.7×10$^{-17}$ | 14.9 | 13.0 | 1.2×10$^3$ | 8.0×10$^4$ | 2 | ∼0 |
| $\alpha$-pinene (R) | 1.0 | 2.6×10$^{-16}$ | 15.1 | 13.5 | 9.9×10$^2$ | 2.7×10$^2$ | ∼0 | ∼0 |
| camphene | 3.1 | 1.2×10$^{-17}$ | 13.7 | 29.3 | 1.2×10$^{-8}$ | ∼0 | ∼0 | ∼0 |
|  |  |  |  | 23.4 | 5.9×10$^{-5}$ | ∼0 | ∼0 | ∼0 |
|  |  |  |  | 40.3 | 6.0×10$^{-18}$ | ∼0 | ∼0 | ∼0 |
| sabinene | 2.0 | 3.4×10$^{-17}$ | 15.6 | 9.7 | 5.0×10$^5$ | 7.0×10$^5$ | 39 | 9 |
| $\alpha$-thujene (R) | 0.2 | 6.5×10$^{-16}$ | 16.1 | 12.0 | 1.4×10$^4$ | 1.2×10$^5$ | 20 | ∼0 |
| $\alpha$-thujene (S) | 1.7 | 2.6×10$^{-17}$ | 17.1 | 12.0 | 1.6×10$^4$ | 2.4×10$^4$ | 7 | ∼0 |
| $\Delta$-carene | 1.4 | 7.4×10$^{-17}$ | 13.7 | × | × | × | × | × |
| limonene | 1.1 | 1.2×10$^{-15}$ | 16.0 | × | × | × | × | × |

difference between the reaction rate coefficients calculated, and thus ring-opening yields, using TST (which does not account for excess energy) and simulations conducted with standard RRKM theory with Eckart tunneling correction ($k_{RRKM}$). This is particularly evident for sabinene, where accounting for excess energy increases the yield of ring-opening from 9% to 39%. Additionally, the calculations show that the ring-opening reaction is significant only for sabinene (39%) and $\alpha$-thujene (up

to 20%). For $\beta$-pinene (2%) it is minor, and for $\alpha$-pinene and camphene, it is negligible. For $\Delta$-carene, we do not predict ring-opening reactions as alkyl radical center is too far from the opening ring structure.





It is worth noting that MESMER-derived rate coefficients tend to be higher than MC-TST rate coefficients as the former does not implicitly account for multiple reactant and TS conformers. A comparison with single-conformer TST equation derived rate coefficients, however, showed that excess energy is still driving the formation of ring opened products. Details are provided

in Supporting Information. The ring-opening reaction is particularly interesting from an oxidation chemistry point of view as it leads to an alkyl radical with less steric constraints than the original "unbroken" counterpart. This promotes more efficient autoxidation and consequently higher molecular functionalization (see Figure 3).

**Figure 3.** Ring-opening radical rearrangement reactions for acetyl–alkyl radicals from bicyclic monoterpenes.

Simulated time profiles of species distributions for bicyclic alkyl radicals reacting via ring–opening versus $O_2$–addition are presented in Figure 4. The results are consistent with the reactions of monoterpenes with $NO_3$ radicals obtained by Draper





et al. (2024). However, ring-opening for APR constitutes a smaller sink for the respective monoterpenes compared to the $NO_3$ case. For example, the ring-opening reaction accounts for 86% in the case of sabinene with the nitrooxy–alkyl radical and 39% for sabinene with the acetyl radical. Ring-opening reaction becomes more competitive as the ring size decreases. For sabinene (39%) and $\alpha$-thujene (20% for R-isomer and 7% for S-isomer), which contain a 3-membered ring, this competition is the greatest, while it is minor for $\beta$-pinene (2%) containing the 4-membered ring and negligible for the 5-membered camphene.

There is also a noticeable difference between sabinene and $\alpha$-thujene, which differ only in the position of the double bond and, consequently, the site of APR addition. This suggests that stereo-electronic effects must influence the reactivity towards ring-opening (Draper et al., 2024).

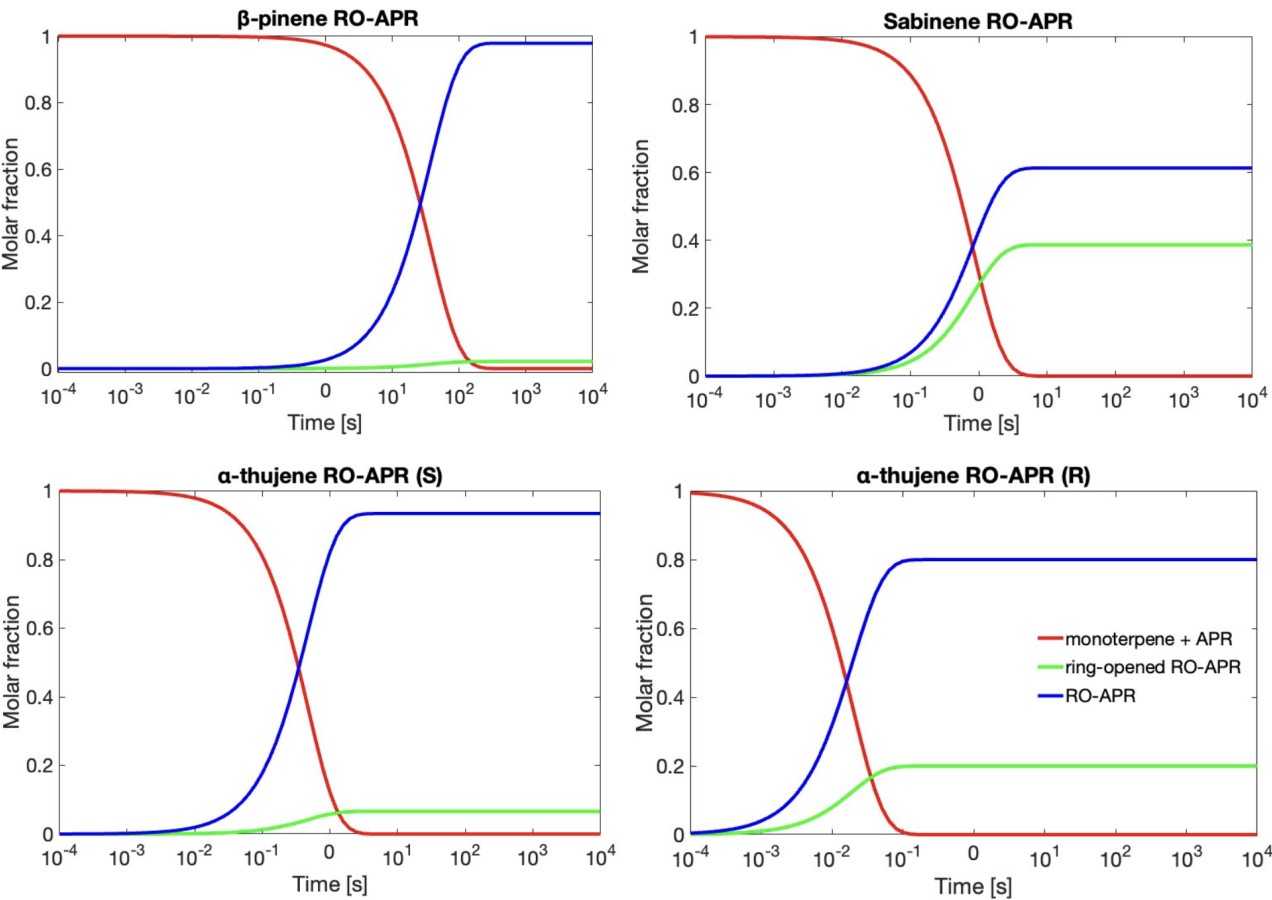

**Figure 4.** RRKM-ME simulated time profiles of species distributions for bicyclic alkyl radicals reacting via ring-opening versus $O_2$-addition.



## 3.2 APR-initiated Autoxidation

The reaction of monoterpenes with APRs results in a compound with a carbon-centered radical, to which $O_2$ can add and
subsequently perform an H-shift reaction. Additionally, for monoterpenes that undergo ring-opening, carbon-centered radicals
are also formed, allowing for further oxidation. To complete the oxidation pathways of the studied monoterpenes, H-shift
reactions for selected intermediate systems were calculated.

Limonene, as it is not a bicyclic monoterpene, cannot undergo alkyl radical ring-opening reactions. Formed in accretion
reaction with APR, the alkyl radical adds oxygen and subsequently it can react via bimolecular reactions, perform $HO_2$ loss or
H-shift reactions. In the case of H-shift reactions, these would result in the formation of another carbon-centered radical. The
considered reactions are illustrated by labels on Figure 5, and corresponding barriers heights and reaction rate coefficients are
presented in Table 2.

**Figure 5.** Labeling for H-shift reactions of limonene peroxyl radical. The numbers on the scheme correspond to the numbering in the Table
2.

**Table 2.** Energy barrier heights [kcal/mol] and calculated TST reaction rate coefficients [$k_{TST}$ 1/s] for studied H-shift reactions of limonene-
derived peroxyl radical. *This rates were calculated using the LC-TST equation (see Supporting Information).

| pathway | barrier | $k_{TST}$ |
|---------|---------|-----------|
| H-shift-1 | 31.5 | $3 \times 10^{-7}$ |
| H-shift-2 | 24.0 | $6 \times 10^{-5}$ |
| H-shift-3 | 24.0 | $1 \times 10^{-3}$ |
| H-shift-4 | 26.3 | $9 \times 10^{-6}$ |
| H-shift-5* | 37.6 | $9 \times 10^{-17}$ |
| H-shift-6* | 30.1 | $1 \times 10^{-10}$ |
| H-shift-7 | 26.6 | $3 \times 10^{-6}$ |

The lowest barriers are exhibited by H-shift-2 (24 kcal/mol) and H-shift-3 (24 kcal/mol). Although the barriers are identical,
the reaction rate coefficient calculated using MC-TST is the highest for H-shift-3, on the order of $10^{-3}$ s$^{-1}$. This difference can
be explained by a significantly higher tunneling factor for this reaction ($\kappa_t$=1192) compared to H-shift-2 ($\kappa_t$=54). This value





is too low for the H-shift reaction to compete with bimolecular reactions that take place in the atmosphere. The calculations presented here draw a direct analogy to the research carried out by Pasik et al. (2024a) concerning the reaction of isoprene with APR radicals. The findings from our current study extend these concepts to limonene. As in the case of isoprene, our results indicate that the H-shift reactions calculated for limonene occur at a rate that is insufficient to drive an autoxidation mechanism.

The slow rate of H-shift reactions implies that alternative pathways (for example with $RO_2$) might be more dominant in the oxidation process of APR + limonene derived $RO_2$.

As mentioned in Section 3.1, the addition of APR to the double bond in sabinene results in the release of energy high enough to lead to secondary ring opening. RRKM-ME calculations show a 39% yield of the alkyl ring-opening reaction in the case of sabinene. As a result, a carbon-centered radical is formed, to which molecular oxygen adds, forming an $RO_2$ radical that could undergo H-shift reactions as indicated in Figure 6.



**Figure 6.** Labeling for H-shift reactions of ring-opened sabinene acetyl–peroxyl radical. The numbers on the scheme correspond to the numbering in the Table 3.


For this system, we tested some possible H-shift reactions, as indicated in Figure 6. Also the 5- and 6-membered ring formation reactions were investigated. The calculated energy barriers and unimolecular reaction rate coefficients are summarized in Table 3 . The data indicate that the fastest reaction occurs for H-shift-4, with a rate constant of $2 \times 10^{-2}$ 1/s. This rate is sufficiently high to compete with bimolecular reactions in a relatively clean atmosphere with low $NO_x$, suggesting that this

mechanism could represent APR-initiated autoxidation. The H-shift reaction leads to a carbon-centered radical capable of adding $O_2$. At this stage, this reaction, along with a competitive ring-opening pathway and subsequent oxidation, produces a compound with 12 carbon atoms and 7 oxygen atoms. Further oxidation can eventually result in a compound with a carbon-to-oxygen ratio near to 1:1.

For the ring closure reactions, the obtained rate coefficients are significantly slower compared to the results from similar

studies Nozière and Vereecken (2024) and Vereecken et al. (2021). In these studies, the 6-membered ring closure proceeds with a rate on the order of $10^{-2}$ 1/s. However, these results refer to unsaturated acyclic $RO_2$ radicals, whereas in the case of sabinene-derived $RO_2$, a constrained ring is present. This may account for the discrepancies observed between the values obtained in these studies. In studies by Møller et al. (2020), the authors investigated H-shift and endoperoxide cyclization for OH derived oxidation products of different monoterpenes. Among these, terpinolene + OH + $O_2$ peroxy radical 5- and



6-membered ring closures were examined, which have similar structure to system we studied. The authors reported rates coefficients for 5-membered endoperoxide cyclization as $10^{-2}$ 1/s. 6-membered ring formation was calculated on a lower level of theory, yielding a rate of $10^{-3}$ 1/s. Comparing our results with those of Møller et al. (2020), the formation of the 5-membered ring is generally faster than that of the 6-membered ring. Differences in calculated rates may be due to different reactant structures.

**Table 3.** Energy barrier heights [kcal/mol] and calculated TST reaction rate coefficients [$k_{TST}$ 1/s] for studies H-shift and ring formation reactions of sabinene-derived peroxyl radical.

| pathway | barrier | $k_{TST}$ |
|---------|---------|-----------|
| H-shift-1 | 34.8 | $3 \times 10^{-13}$ |
| H-shift-2 | 29.7 | $4 \times 10^{-6}$ |
| H-shift-3 | 25.1 | $1 \times 10^{-5}$ |
| H-shift-4 | 22.8 | $2 \times 10^{-2}$ |
| H-shift-5 | 41.7 | $2 \times 10^{-13}$ |
| 5-ring | 19.9 | $3 \times 10^{-3}$ |
| 6-ring | 22.6 | $3 \times 10^{-5}$ |

## 210   3.3   Alkoxy Radical $\beta$-scission Reactions

Competitively to H-shifts, formed peroxy radical can undergo bimolecular reactions with radicals present in the atmosphere like $RO_2$ or $HO_2$ to form alkoxy radical and subsequently undergo $\beta$-scission reactions (see Figure 2 for overview). For endocyclic monoterpenes, both "right" (R$\beta$1) and "left" (R$\beta$2) $\beta$-scissions were considered, whereas for exocyclic monoterpenes, "right", "left", and "top" (R$\beta$3) C−C bond scissions were analyzed (see Figure 7).

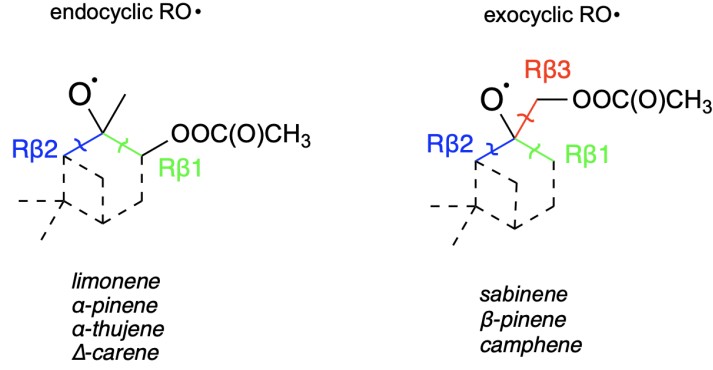

**Figure 7.** Reaction labeling scheme for alkoxy scissions reactions.





The stereochemistry was included according to Figure 8. Table 4 presents the calculated barriers for the corresponding C−C
bond scissions, MC-TST reaction rate coefficients, and product yields for the reactions for seven monoterpenes.

**Figure 8.** Molecular structures of examined acetyl-alkoxyl stereoisomers of limonene, $\alpha$-thujene, sabinene, $\beta$-pinene, $\Delta$-carene, $\alpha$-pinene and camphene.

From the data we can draw general conclusions: for all monoterpenes, C−C bond scission reactions that result in a radical center on a 3-, 4-, or 5-membered ring structure exhibit the highest energy barriers. These findings align with the results reported by Draper et al. (2024) throughout for nitrooxy alkoxyl $\beta$-scission reactions. Interestingly, for the exocyclic monoterpenes

sabinene and $\beta$-pinene, the preference for the $\beta$-scission pathway significantly depends on stereochemistry. This is particularly evident for sabinene, where the R isomer predominantly follows the R$\beta$3 pathway, whereas the S isomer prefers the R$\beta$1 pathway. For $\beta$-pinene, both isomers predominantly undergo a reaction leading to a closed-shell ketone (R$\beta$3), but this is more preferred in the R-alkoxyl isomer (88%) compared to the S-alkoxyl isomer (53%). For camphene, which is also an exocyclic



**Table 4.** Zero-point corrected barrier heights ($\Delta E$) for considered acetyl–alkoxy $\beta$-scission reactions, corresponding MC-TST rate coefficients and product yields.

| monoterpene | $\Delta E$ | [kcal/mol] | | $k_{TST}$ | [1/s] | | Yield [%] | | |
|---|---|---|---|---|---|---|---|---|---|
| | R$\beta$1 | R$\beta$2 | R$\beta$3 | R$\beta$1 | R$\beta$2 | R$\beta$3 | R$\beta$1 | R$\beta$2 | R$\beta$3 |
| $\beta$-pinene(R) | 10.7 | 12.0 | 8.1 | $5\times10^4$ | $4.6\times10^3$ | $4\times10^5$ | 11 | 1 | 88 |
| $\beta$-pinene (S) | 9.9 | 12.4 | 9.7 | $1.5\times10^5$ | $3.7\times10^3$ | $1.7\times10^5$ | 46 | 1 | 53 |
| camphene(R) | 4.5 | 1.5 | 8.6 | $2.1\times10^{11}$ | $3.7\times10^8$ | $5.1\times10^5$ | 100 | 0 | 0 |
| camphene (S) | 5.5 | 2.6 | 11.1 | $3.7\times10^{10}$ | $1.9\times10^8$ | $9.3\times10^3$ | 99 | 1 | 0 |
| sabinene (R) | 10.6 | 15.6 | 9.6 | $1.5\times10^5$ | $5.1\times10^1$ | $2.2\times10^5$ | 37 | 0 | 63 |
| sabinene (S) | 9.6 | 15.8 | 8.7 | $8.8\times10^5$ | 8.35 | $4\times10^5$ | 69 | 0 | 31 |
| $\alpha$-thujene (RR) | 6.5 | 15.9 | $\times$ | $2.0\times10^8$ | $2.1\times10^1$ | $\times$ | 100 | 0 | $\times$ |
| $\alpha$-thujene (RS) | 4.4 | 14.4 | $\times$ | $2.5\times10^9$ | $4.9\times10^2$ | $\times$ | 100 | 0 | $\times$ |
| $\alpha$-thujene (SR) | 4.5 | 14.9 | $\times$ | $2.3\times10^9$ | $1.0\times10^2$ | $\times$ | 100 | 0 | $\times$ |
| $\alpha$-thujene (SS) | 6.6 | 15.3 | $\times$ | $6.0\times10^7$ | $5.5\times10^1$ | $\times$ | 100 | 0 | $\times$ |
| limonene (RR) | 5.9 | 10.4 | $\times$ | $5.3\times10^7$ | $4.9\times10^4$ | $\times$ | 100 | 0 | $\times$ |
| limonene (RS) | 6.6 | 10.9 | $\times$ | $2.1\times10^7$ | $5.0\times10^4$ | $\times$ | 100 | 0 | $\times$ |
| limonene (SR) | 6.7 | 10.8 | $\times$ | $7.3\times10^7$ | $3.7\times10^4$ | $\times$ | 100 | 0 | $\times$ |
| limonene (SS) | 5.9 | 8.4 | $\times$ | $2.5\times10^8$ | $3.5\times10^6$ | $\times$ | 99 | 1 | $\times$ |
| $\Delta$-carene (RR) | 6.5 | 9.4 | $\times$ | $1.6\times10^8$ | $5.7\times10^6$ | $\times$ | 96 | 4 | $\times$ |
| $\Delta$-carene (RS) | 6.9 | 8.7 | $\times$ | $3.5\times10^7$ | $3.8\times10^6$ | $\times$ | 90 | 10 | $\times$ |
| $\Delta$-carene (SR) | 7.7 | 8.5 | $\times$ | $9.8\times10^6$ | $3.2\times10^6$ | $\times$ | 75 | 25 | $\times$ |
| $\Delta$-carene (SS) | 7.1 | 7.4 | $\times$ | $1.2\times10^8$ | $2.0\times10^7$ | $\times$ | 86 | 14 | $\times$ |
| $\alpha$-pinene (RR) | 7.7 | 11.7 | $\times$ | $4.7\times10^6$ | $1.4\times10^4$ | $\times$ | 100 | 0 | $\times$ |
| $\alpha$-pinene (RS) | 6.3 | 11.6 | $\times$ | $7.9\times10^7$ | $1.8\times10^4$ | $\times$ | 100 | 0 | $\times$ |
| $\alpha$-pinene (SR) | 4.5 | 11.6 | $\times$ | $4.2\times10^8$ | $3.4\times10^4$ | $\times$ | 100 | 0 | $\times$ |
| $\alpha$-pinene (SS) | 6.7 | 11.6 | $\times$ | $1.7\times10^7$ | $2.7\times10^4$ | $\times$ | 100 | 0 | $\times$ |





monoterpene, a clear preference for right-side C−C bond cleavage (Rβ1) is observed. For α-pinene, α-thujene, and limonene,
regardless of stereoisomerism, a pronounced major product of Rβ1 C−C bond scission is consistently observed. This pathway
leads to termination of the oxidative chain reaction. In contrast to the other investigated endocyclic monoterpenes, Δ-carene
exhibits competitive pathways, with each stereoisomer showing its own preference.

Figures 9–15 present the proposed fate for each studied monoterpene. Despite their structural similarities and belonging to
the same group of compounds, each studied compound exhibits its own unique reactivity. Consequently, the potential contri-
bution to the formation and growth of organic aerosols will differ for each monoterpene. This underscores the importance of
these studies, as differences that arise early in the oxidation pathways can significantly impact the overall outcomes.

### 3.3.1 Limonene

Figure 9 presents the proposed fate in the APR-initiated oxidation pathway of limonene. Since limonene does not have a
bicyclic structure, it is incapable of undergoing alkyl radical ring-opening reactions. Therefore, after the accretion reaction
with APR, where an alkyl radical is formed, it most likely undergoes $O_2$ addition. As shown in section 3.2, H-shift reactions
for this intermediate are too slow, making it most likely to react bimolecularly to form an alkoxy radical. Further calculations
demonstrate a clear dominance of C−C bond scissions following the Rβ1 pathway, leading to the α-QOOR alkyl radical.
Multiple studies indicate that α-QOOR radicals are unstable and undergo decomposition (Vereecken et al., 2004; Anglada
et al., 2016; Rissanen et al., 2014). In the case of limonene, this leads to the detachment of the $CH_3C(O)O\cdot$ radical and the
formation of closed-shell endolim. This signifies the termination of oxidation chain, although the formed endolim has a C=C
double bond capable of reacting in a second-generation oxidation pathway. A relatively simple and unbranched first-generation
oxidation pathway would most likely lead to relatively small SOA yields in APR-initiated oxidation. This is only a hypothesis
and further research is needed to confirm the exact mechanisms and factors contributing to the SOA yields.

### 3.3.2 α-pinene

Next, we propose the oxidation mechanism for α-pinene (see Figure 10). As mentioned earlier, the in-depth analysis of the
addition of $CH_3C(O)OO\cdot$ to the double bond in α-pinene revealed a new, lower barrier compared to Pasik et al. (2024a) that
proceeds through tertiary radical formation pathway. This finding arises from considering the side from which the radical
attacks the double bond. The analysis shows that the lowest barrier occurs through the formation of a tertiary radical in the
R-stereoisomer of α-pinene + APR product.

The opening of the secondary ring rearrangement was considered. Calculations show that about 100% of the accretion
product undergoes $O_2$ addition, and ring opening does not occur. Assuming the formation of an alkoxy radical in the subsequent
step, both right and left C−C bond scissions were examined. Similar studies of α-pinene + $NO_3$ were conducted by Kurtén
et al. (2017) and subsequently reviewed by Draper et al. (2024). According to their calculations, the reaction between α-
pinene and $NO_3$ proceeds with the formation of a tertiary radical, and nitrooxy alkoxy β-scissions lead to the main product,
α-pinoaldehyde, through left-side C−C bond scission. TST reaction rate coefficient calculations show that for all stereoisomers
of α-pinene-derived RO radicals, the main product is α-pinoaldehyde formed via Rβ1. Similar to the findings of Kurtén et al.





**Figure 9.** Proposed fate of limonene + $CH_3C(O)OO\cdot$. Dashed box present closed-shell product that terminates oxidation pathway.

(2017), the oxidation pathway of $\alpha$-pinene leads to a closed-shell compound, resulting in the termination of the oxidation chain. This would lead to the small secondary organic aerosol (SOA) yields of $\alpha$-pinene.

### 3.3.3 $\beta$-pinene

We now discuss the oxidation pathway initiated by APR for $\beta$-pinene, as shown in Figure 11. Calculations indicate that the alkyl radical ring-opening rearrangement accounts for a minor yield of 2%. This means that nearly all alkyl radicals proceed without ring opening and undergo $O_2$ addition, leading to further reactions that result in the formation of the alkoxy radical. $\beta$-Pinene is unique among exocyclic monoterpenes in that it provides significant branching ratios for all three possible C−C bond scissions. The most favored pathway is R$\beta$3 for both stereoisomers, but with different yields — 88% for the R-alkoxy isomer

and significantly less, 53%, for the S-alkoxy isomer. This reaction leads to the formation of closed-shell nopinone. Similarly, for both stereoisomers, the least favored pathway is the one leading to the left side scission (R$\beta$2). Despite its similarity to $\alpha$-pinene, $\beta$-pinene has the potential for various C−C bond scissions, whereas $\alpha$-pinene favors the pathway leading to the closed-shell $\alpha$-pinonaldehyde. Although for $\beta$-pinene, the analogous pathway leading to nopinone is the most favored, other pathways are also competitive. This would result in the higher SOA yields observed for $\beta$-pinene compared to $\alpha$-pinene.





**Figure 10.** Proposed fate of $\alpha$-pinene + CH$_3$C(O)OO·. Dashed box present closed-shell product that terminates oxidation pathway.

**Figure 11.** Proposed fate of $\beta$-pinene + CH$_3$C(O)OO·. Dashed box present closed-shell product that terminates oxidation pathway. The indicated yields for $\beta$-scission reactions depend on the stereoisomer (see Table 4).



### 3.3.4 Camphene

The mechanism for camphene APR-initiated oxidation is shown in Figure 12. Due to the presence of a 5-membered ring in its structure, the ring-opening reaction contributes to smaller strain relief than in smaller-sized rings, directly resulting in a high energy barrier for this reaction. Consequently, ring-opening reactions are not competitive with $O_2$ addition. For alkoxy $\beta$-scissions, the calculated reaction rate coefficients indicate that the vast majority (99–100%) of camphene intermediates will favor right-sided scission (R$\beta$1), leading to a carbon-centered radical capable of further oxidation. The minor pathway, R$\beta$2, also leads to an intermediate whose oxidation pathway does not terminate early. It is noteworthy that the calculated reaction rate coefficients for both $\beta$-scissions of camphene radicals are faster than those for any other monoterpenes studied here. This is consistent with the results obtained by Draper et al. (2024). Moreover, the reaction leading to the formation of closed-shell camphenilone (R$\beta$3) is negligible, indicating no early oxidative chain termination and a potential for high SOA yields from camphene.

**Figure 12.** Proposed fate of camphene + $CH_3C(O)OO\cdot$. Dashed box present closed-shell product that terminates oxidation pathway.

### 3.3.5 $\alpha$-thujene

The mechanism for $\alpha$-thujene oxidation is shown in Figure 13. Due to the presence of a 3-membered ring, the additional ring strain is significant enough that the earlier alkyl ring-opening radical rearrangement reaction becomes competitive with $O_2$ addition, with a small but not negligible yield of 3–11% (depending on the stereoisomer). The ring-opening reaction allows for an additional branch in the oxidation pathway of $\alpha$-thujene. The resulting rearranged radical most likely undergoes $O_2$ addition, forming a peroxy radical with the potential for further propagation reactions via H-shifts, cyclization, bimolecular reactions or addition to the newly formed double bond. Besides unimolecular ring-opening rearrangement, the dominant pathway pro-



ceeds through $O_2$ addition and the formation of an monoterpene + APR- derived alkoxy radical. For the two possible alkoxy scissions, we observe the exclusive formation of the R$\beta$1 product regardless of the stereoisomer. This results in the formation

of an $\alpha$-QOOR alkyl radical, which decomposes to $\alpha$-thujonaldehyde providing an additional source of this compound in the atmosphere.

**Figure 13.** Proposed fate of $\alpha$-thujene + $CH_3C(O)OO\cdot$. Dashed box present closed-shell product that terminates oxidation pathway.

### 3.3.6  Sabinene

The most interesting case is represented by sabinene. Among all bicyclic monoterpenes investigated, it was found that this monoterpene undergoes a competitive reaction with alkyl radical ring opening with a yield of 39% compared to direct $O_2$

addition. This is significant enough to impact the chemistry of this compound and its oxidation pathways. Calculations show that the radical formed in this reaction can add $O_2$ and perform further H-shift reaction with rate constant $2\times10^{-2}$ 1/s leading to more oxidized compound able to undergo additional propagation steps (via $RO_2$ bimolecular reactions or unimolecular reaction) leading to highly oxidized compound with potential to participate in SOA formation.

Regarding the further oxidation of the 61% that proceeds without ring opening, the most favorable pathway for the alkoxy

radical channels differs for the two RO-APR stereoisomers. The R-alkoxy favors the R$\beta$3 pathway, leading to closed-shell sabinaketone (63%) and termination of the oxidative chain. On the other hand, the majority of the S-alkoxy isomer favors the R$\beta$1 scission (69%), leading to a radical capable of further propagation. Additionally, the left side $\beta$-scissions showed 0% yield for both stereoisomers.





**Figure 14.** Proposed fate of sabinene + CH$_3$C(O)OO·. Dashed box present closed-shell product that terminates oxidation pathway. The indicated yields for $\beta$-scission reactions depend on the stereoisomer (see Table 4)

For the right-side reaction, which facilitates further propagation, H-shift reactions for the RO$_2$ radical were investigated (see
SI for overview). The fastest of these reactions proceeds with a rate constant of $1.3 \times 10^{-1}$ s$^{-1}$, sufficient to compete with bimolecular reactions or alkyl radical 3-membered ring opening. Moreover, this reaction produces another radical that can undergo further oxidation. Along with alkyl ring-opening rearrangement, this suggests high SOA yields from sabinene + APR.

### 3.3.7   Δ-carene

The proposed oxidation pathway for Δ-carene is shown in Figure 15. Δ-carene exhibits a unique oxidation pathway. For the
two possible C−C bond scissions, Δ-carene most prefers the right-side alkoxy $\beta$-scission, but with varying branching ratios for different stereoisomers. This contrasts with the NO$_3$-initiated oxidation chain published by Kurtén et al. (2017) and later Draper et al. (2024), where their calculations showed that Δ-carene prefers left-side alkoxy scissions. This confirms the uniqueness of Δ-carene among monoterpenes and highlights the necessity for further investigation into its distinct oxidation behavior. In the case of R$\beta$1 bond scission, an $\alpha$-QOOR alkyl radical is formed, which then decomposes to caronaldehyde. The less
preferred but still possible left-side C−C bond scission leads to the formation of a carbon-centered radical, which can undergo further secondary ring-opening reactions or O$_2$ addition. This opens an oxidation pathway for Δ-carene and would lead to the relatively high SOA yields.



**Figure 15.** Proposed fate of Δ-carene + CH$_3$C(O)OO·. Dashed box present closed-shell product that terminates oxidation pathway.

## 4 Conclusions

In this study, we investigated the possible oxidation pathways for APR-initiated oxidation of seven atmospherically relevant
monoterpenes. Building upon research by Pasik et al. (2024a), Draper et al. (2024), and Kurtén et al. (2017), we identified
oxidation pathways that could lead the high SOA yields for certain monoterpenes and low SOA yields for others.

Our calculations indicate that alkyl radical rearrangement is significant primarily for sabinene and $\alpha$-thujene, where this
pathway competes with O$_2$ addition. For the other monoterpenes ($\beta$-pinene, $\alpha$-pinene, camphene), this channel is minor (0–
1%). This is consistent with observations by Draper et al. (2024), who noted that the preference for ring opening is partly
dependent on ring size and the resulting ring strain released—this effect being more pronounced for smaller rings. The variance
in ring-opening yields seen in sabinene and $\alpha$-thujene, both featuring a 3-membered ring, implies that stereoelectronic effects
and the APR's position may significantly influence reactivity. For sabinene, further O$_2$ addition and H-shift reactions were
investigated. The fastest H-shift, which leads to resonance stabilized allylic radical, has a reaction rate coefficient on the order
of 0.01 1/s, making it competitive under atmospheric conditions. This oxidation pathway may represent an APR-initiated
autoxidation mechanism proposed by Pasik et al. (2024a).

Our calculations show that each monoterpene has its unique reactivity towards oxidation pathways. The probable oxidation
pathway for limonene leads to the formation of closed-shell endolim, which terminates the oxidation chain. This would lead
to low SOA yield of APR-initiated oxidation of limonene, as the termination prevents further radical propagation and aerosol





formation. Despite having similar structures, the $\beta$-scission rearrangements for $\alpha$- and $\beta$-pinene differ significantly due to the

position of the double bond. For $\alpha$-pinene, the most favorable pathway leads to the formation of an $\alpha$-QOOR radical that decomposes to $\alpha$-pinonaldehyde. In contrast, for $\beta$-pinene, this pathway competes with C−C bond scission that allows for further radical propagation steps. For camphene with a 5-membered secondary ring, alkoxy rearrangements predominantly lead to right-sided scission (R$\beta$1), resulting in a carbon-centered radical capable of further oxidation. In the case of $\alpha$-thujene and sabinene, which differ only in the position of the double bond, significant differences in preferred oxidation pathways

are observed. While $\alpha$-thujene consistently undergoes complete right-side $\beta$-scission leading ultimately to $\alpha$-thujonaldehyde, sabinene exhibits two competitive pathways with varying stereochemical preferences. These intriguing observations indicate that steric effects and stereochemistry clearly play a significant role in controlling oxidative chains. For $\Delta$-carene, major preference for right-side $\beta$-scission is observed, varying between 75% to 96% depending on the stereoisomer. Results suggest a general conclusion that all endocyclic monoterpenes prefer right-side C−C bond scission leading to closed-shell products.

However, exocyclic monoterpenes do not exhibit a consistent trend, but their oxidation pathways lead through intermediate products allowing for further propagation steps and higher SOA contribution.

This work highlights the importance of APRs in atmospheric oxidation chemistry, particularly in the formation of oxygenated multifunctional compounds. However, it should be noted that although the rearrangements presented above exhibit feasible rate constants under atmospheric conditions, the overall significance of these reactions is highly dependent on the initial step,

which is the addition of APR to the double bond in a monoterpene. While calculations may indicate a small percentage compared to OH oxidation, the actual results are heavily dependent on the real concentrations of APRs in the atmosphere and the actual reaction rate constants. Furthermore, as shown by multiple studies, the main sink for APR is its reaction with NOx to form PANs. Consequently, the reactions of APRs with monoterpenes (or unsaturated hydrocarbons in general) will be most significant in areas with low concentrations of NOx (Orlando and Tyndall, 2012; Zheng et al., 2011; Cleary et al., 2007; Liu

et al., 2021; Rissanen, 2018). With this work, authors aim to emphasize the importance of further research in this direction, particularly experimental studies, to determine whether APR-initiated oxidation can indeed impact the atmospheric chemistry of unsaturated hydrocarbons.





*Data availability.* The optimized structures and calculation output files of all relevant compounds that support the findings of the manuscript will be available in the Zenodo repository.

*Author contributions.* DP performed the calculations and wrote the manuscript, TGA assisted with the MESMER calculations, EA carried out the calculations for Section 3.2 for sabinene, and TGA, SI, and NM contributed to the analysis. The study was designed and supervised by NM. All authors proofread the manuscript.

*Competing interests.* The corresponding author has stated that none of the authors have any competing interests.

*Disclaimer.*

*Acknowledgements.* We acknowledge Research Council of Finland for funding (grant nos 347775 and 355966) and the CSC-IT Center for Science in Espoo, Finland, for computational resources.

Dominika thanks the Doctoral Programme in Atmospheric Sciences (ATM-DP) at the University of Helsinki for provided funding.



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
