# Peer review of "Monoterpene oxidation pathways initiated by acyl peroxy radical addition"

_EGUsphere, 2024_

## Author Response (AR1)

**Referee: 1. Comments to the Author**

In this manuscript, the authors provide quantum chemical and theoretical kinetics data that demonstrate the diversity of monoterpene + acylperoxy radical reactions. The authors present a very clear exposition of the ring-opening, hydrogen-shift, and beta-scission reactions to expect for each of seven terpenes considered in the manuscript. The manuscript makes frequent, relevant references to previous mechanistic studies and provides solid evidence for which terpenes may be expected to generate significant secondary organic aerosol due to initial attack by acylperoxy. This is a valuable contribution to the atmospheric chemistry literature.

I have three minor scientific issues for the authors to consider. First, as presented in Table 1, the rates of ring-opening rearrangements as predicted by RRKM theory are higher than those predicted by transition state theory for beta-pinene, sabinene, and alpha-thujene. Clearly, then, these acylperoxy-terpene adducts are chemically activated. Thus, there is the possibility that the yields of ring-opened products, as depicted in Figure 4, may be pressure-dependent. The authors should address this possibility. Second, on p. 11, the authors compare the rates of 5-membered-ring vs. 6-membered-ring endoperoxide cyclizations. The fact that the 5-membered-ring process involves the formation of a presumably more stable tertiary carbon-centered radical should be considered as a reason for the lower activation barrier. Finally, the focus on the conversion of peroxy radicals to alkoxy radicals, starting on p. 11, is understandable and justified, but it would be good to acknowledge the competing pathway of $RO_2 + HO_2 \rightarrow ROOH + O_2$.

Technical corrections: (1) In Figure 9, the rightmost structure should have two C=O double bonds, but the C=O next to the radical center is hard to discern. (2) In Figure 14, the rightmost block of text is almost too small to be legible.

We thank the reviewer for their comments and valuable insights. The manuscript has been revised to address the points raised, which have clarified the study.

**Regarding the first comment (pressure dependence):**

We fully agree with this observation. We recalculated MESMER ring-opening yields for one monoterpene system (sabinene) in four different pressures and put the results in SI. The following has been added to the manuscript:

"It is worth noting that the ring-opening reactions are pressure dependent, with the ring-opened product yield increasing with decreasing pressure for the sabinene system, indicating that the acetyl peroxy-terpene adducts are chemically activated (see SI, Table S5)."

**Regarding the second comment (tertiary carbon-centered radical):**

We also agree with the observation regarding the formation of a tertiary carbon-centered radical in endocyclization reactions as a reason for the lower barrier compared to the other reaction described on page 11. The following has been added to the manuscript:

"Comparing our results with those of \cite{moller2020double}, the formation of the 5-membered ring is generally faster than that of the 6-membered ring, which can be attributed to the formation of a stabilized tertiary carbon-centered radical in the 5-membered case."

**Lastly, we have updated the manuscript in Section 3.3 to include the following:** "Reaction with or \ce{HO2} could also lead to the formation of hydroperoxides and molecular oxygen, following \ce{RO2} + \ce{HO2} \rightarrow \ce{ROOH} + \ce{O2}. However, this process was not investigated in this study."

Technical corrections:

(1) In Figure 9, the rightmost structure should have two C=O double bonds, but the C=O next to the radical center is hard to discern.

This was corrected.

(2) In Figure 14, the rightmost block of text is almost too small to be legible.

This was corrected.

We thank the reviewer for pointing out the technical errors, which have been corrected. Modifications made to manuscript are visible in tex file.

**Referee: 2. Comments to the Author**

The manuscript describes a theoretical study of the very complex reaction mechanisms arising from addition of the $CH_3C(O)OO$ radical (APR) to in total seven different monoterpenes (MT) in presence of $O_2$. Particular emphasis is laid on the kinetics of the competition between unimolecular reactions (bond dissociation, hydrogen shift, and ring opening) of the intermediarily formed alkyl-type radicals and their addition of $O_2$. The work seems in part to supplement and extend an experimental study published very recently by three of the authors jointly with three other researchers (Pasik 2024a). In the current study, advanced quantum chemical methods were used to calculate molecular

and transition state properties, and statistical rate theory was applied to derive rate constants and branching ratios. In general, all the methods used are state-of-the-art and appear adequate. The mechanisms derived and the calculated kinetic data are discussed in great detail, including even stereochemical aspects. Conclusions about the atmospheric relevance of these oxidation pathways are also discussed.

So, all in all, this is a very laborious work exploring a special branch of atmospheric monoterpene oxidation. Whether this branch is really important has still to be shown and depends essentially on the actual atmospheric APR concentration, a point which is also mentioned by the authors. Because also important general mechanistic and kinetic information on atmospheric terpene oxidation is provided, the manuscript can generally be recommended for publication in ACP.

There is however one important point that should be addressed by the authors before final acceptance: The **presentation of the kinetics calculations** should be improved. From the current version of the manuscript, it is very difficult to detect, which method was applied to which reaction step: Obviously, the bimolecular reaction APR + MT was characterized by MC-TST, eq. (1). The rate constant of the subsequent ring-opening reaction was calculated by both TST-2, eq. (2), and RRKM (not specified, probably within MESMER). Here the APR-addition reaction (APR + MT?) was simulated using ILT (line 108). Does this rate coefficient grossly agree with the value obtained from eq. (1)? What was the rationale for choosing $1 \times 10^{-12}$ cm$^3$ molecule$^{-1}$ s$^{-1}$ for $O_2$ addition? What does $k_{TST-1}$ (Table 1) characterize: APR + MT or alkyl + O2 or ring-opening product + $O_2$? This is difficult to find out. It is reasonable that $k_{RRKM}$ (should it not better read $k_{MESMER}$?) in Table 1 is larger than $k_{TST-2}$. But this is not always the case, why? In Figure 4, species profiles are plotted; what is the difference between RO-APR and ring-opened RO-APR?

All these points (which can also not be resolved by looking into the supplemental material or into reference Pasik 2024a) should be presented a bit more clearly and more conclusively, maybe within a graphical scheme. This would help the reader of this otherwise convincing paper very much.

Some minor, more formal points are:

general: The authors should consider using the term 'acetyl peroxyl" instead of 'acyl peroxyl' because the latter may suggest RC(O)OO whereas only $CH_3C(O)OO$ was studied in the present work.

line 37: 'unique' should probably better read 'specific'

line 44: insert [X] '... and [this] reaction ... to lead [to] epoxides ...'

line 45 and elsewhere: for elementary reactions, the term 'accretion' is very unusual, it should better read 'association' or 'addition'

line 73: What was done with 'squared rotational constant values'?

line 86: Different conformers of the acyl peroxyl radical were accounted for. What about possible conformers of the monoterpenes (alpha-thujene, sabinene)?

line 104: 'reaction dynamics' should read 'reaction kinetics'

Figure 2: please check the meaning of $RO_2$ above the third arrow.

We thank you for your comments and suggestions, which we have addressed with the following revisions.

Reviewer comment: "[...] Obviously, the bimolecular reaction APR + MT was characterized by MC-TST, eq. (1). The rate constant of the subsequent ring-opening reaction was calculated by both TST-2, eq. (2), and RRKM (not specified, probably within MESMER)."

Regarding the **presentation of kinetic data,** the manuscript has been modified to clarify the methodology used. Described variables have been added in brackets at the end of relevant descriptions to help readers follow the text more easily, and these updates have also been incorporated into the workflow figure.

"Here the APR-addition reaction (APR + MT?) was simulated using ILT (line 108). Does this rate coefficient grossly agree with the value obtained from eq. (1)?"

The APR + MT rate coefficients were only calculated using equation (1). These rate coefficients were later used as pre-Arrhenius factors in the MesmerILT method to treat APR + MT association reactions is MESMER. We have now made this clear in the manuscript.

Following changes to the manuscript has been made:

"[...] the monoterpene+APR-addition reaction was simulated using the using the Mesmer Inverse Laplace Transform (MesmerILT) method, assuming a barrierless association and using realistic bimolecular rate coefficients calculated using eq. (1) (**k\$_{TST-1}\$ was used as a pre-exponential factor in ILT reactions**)."

"What was the rationale for choosing $1 \times 10^{-12}$ $cm^3$ $molecule^{-1}$ $s^{-1}$ for $O_2$ addition?"

Thank you for pointing this out. This value was incorrect and has now been changed to $6 \times 10^{-12}$ $cm^3$ $molecule^{-1}$ $s^{-1}$ for $O_2$ addition, a value that has been used previously in similar studies. References to these studies have also been added. Moreover, the branching ratios for the studied reactions were recalculated using the correct $O_2$ addition rate, and the changes have been applied to the manuscript. The most significant change was observed in the branching ratio for the ring-opening reaction of sabinene, which decreased from 39% to 31%. This adjustment has been incorporated into the text and

updated in the relevant figures (Figure 4, Figure 13, Figure 14). The following sentence has been added to the manuscript:

These values have previously been used for \ce{O2}-addition reactions to carbon-centered radicals \cite{draper2024unpacking, berndt2015gas, kurtén2017alkoxy}.

"What does $k_{TST-1}$ (Table 1) characterize: APR + MT or alkyl + O2 or ring-opening product + $O_2$?"

$k_{TST-1}$ characterizes the addition of the APR to the monoterpene (MT). To make this clearer, we have modified Figure 2 by adding the abbreviations next to the corresponding arrow/reaction and in manuscript text.

It is reasonable that $k_{RRKM}$ (should it not better read $k_{MESMER}$?) in Table 1 is larger than $k_{TST-2}$. But this is not always the case, why?

We now use $k_{MESMER}$ instead of $k_{RRKM}$ in the manuscript. The MC-TST rate calculated for $\alpha$-pinene is larger than the $k_{MESMER}$ value due to the identification of more $\alpha$-pinene-TS conformers, which resulted in a larger partition function contribution. In contrast, the LC-TST rate (details provided in the SI, Table S2) is smaller than the $k_{MESMER}$ value, aligning with expectations.

In Figure 4, species profiles are plotted; what is the difference between RO-APR and ring-opened RO-APR?

Thank you for this comment. It was unclear so we changed RO-APR for R-APR, which represents monoterpene+APR accretion product (and also substrate for ring-opening reaction). Ring-opened R-APR($O_2$) represents product of $O_2$-addition to ring-opened monoterpene+APR system.

Additionally, we added color coding in the Figure 4 caption for clarity.

Figure 4 caption has been modified as follows: The red line represents the reactant (monoterpene+CH3C(O)OO.) for ring-opening reaction, the green line corresponds to products of \ce{O2}-addition to the intact alkyl radical (R-APR($O_2$) and the blue line represents the remaining non-ring-opened product that undergoes further oxidation with $O_2$.}

general: The authors should consider using the term 'acetyl peroxyl" instead of 'acyl peroxyl' because the latter may suggest RC(O)OO whereas only $CH_3C(O)OO$ was studied in the present work.

We agree. The manuscript text now specifies acetyl peroxy radical instead of the more general acyl peroxy radical.

line 37: 'unique' should probably better read 'specific'

This was changed.

line 44: insert [X] '... and [this] reaction ... to lead [to] epoxides ...'

This was changed.

line 45 and elsewhere: for elementary reactions, the term 'accretion' is very unusual, it should better read 'association' or 'addition'

Regarding the use of the term "accretion," we have decided to retain it for consistency with our previous publications.

line 73: What was done with 'squared rotational constant values'?

To address the identification of duplicate conformers, squared rotational constant values, along with electronic energy and dipole moments, were used to differentiate and identify duplicates, which were subsequently removed. These variables were chosen because, together, they provide reliable identification of duplicates.

line 86: Different conformers of the acyl peroxyl radical were accounted for. What about possible conformers of the monoterpenes (alpha-thujene, sabinene)?

Thank you for this comment. The MC-TST equation was modified to account for different conformers of monoterpene reactants in the partition function.

Changes to manuscript:

Equation 1 has been changed to account for different conformers of monoterpene reactants in the partition function.

Following has been added: "$Q_{\mathrm{MT}}$ is the partition function of the monoterpene. Most monoterpenes only have one conformer. For sabinene and $\alpha$-thujene we found three conformers each and all three were included in the equation."

line 104: 'reaction dynamics' should read 'reaction kinetics'

This was changed.

Figure 2: please check the meaning of $RO_2$ above the third arrow.

This was corrected.

We thank you for pointing this out, and we hope that these revisions make the paper clearer and easier to follow. Modifications made to manuscript are visible in tex file.